# Effective Targeting of Glutamine Synthetase with Amino Acid Analogs as a Novel Therapeutic Approach in Breast Cancer

**DOI:** 10.3390/ijms26010078

**Published:** 2024-12-25

**Authors:** Shimaa Abdelsattar, Hiba S. Al-Amodi, Hala F. Kamel, Ahood A. Al-Eidan, Marwa M. Mahfouz, Kareem El khashab, Amany M. Elshamy, Mohamed S. Basiouny, Mohamed A. Khalil, Khaled A. Elawdan, Shorouk Elsaka, Salwa E. Mohamed, Hany Khalil

**Affiliations:** 1Clinical Biochemistry and Molecular Diagnostics Department, National Liver Institute, Menoufia University, Shebin El-Kom 32511, Egypt; shimaa.abdelsattar@liver.menofia.edu.eg; 2Biochemistry Department, Faculty of Medicine, Umm Al-Qura University, Makkah 21955, Saudi Arabia; hsamodi@uqu.edu.sa (H.S.A.-A.); hfkamel@uqu.edu.sa (H.F.K.); 3Medical Biochemistry and Molecular Biology Department, Faculty of Medicine, Ain Shams University, Cairo 11591, Egypt; 4Department of Biology, College of Science, Imam Abdulrahman Bin Faisal University, Dammam 34212, Saudi Arabia; aeidan@iau.edu.sa; 5Department of Pharmacology and Toxicology, Faculty of Pharmacy, Menoufia University, Shebin El-Kom 32511, Egypt; marwamahfouz111@gmail.com; 6Medical Laboratory Department, High Technology Institute of Applied Health Science, Badr Academy for Science and Technology, Badr City 11829, Egypt; kemo28597@gmail.com; 7Medical Laboratory Science Department, School of Allied Health Sciences, Badr University in Cairo, Badr City 11829, Egypt; 8School of Biotechnology, Badr University in Cairo, Badr City 11829, Egypt; mohamed-salah@buc.edu.eg; 9Clinical Pathology Department, National Cancer Institute, Cairo University, Giza 12613, Egypt; mohamed.ahmedahmed@nci.cu.edu.eg; 10Department of Molecular Biology, Genetic Engineering and Biotechnology Research Institute, University of Sadat City, Sadat City 32897, Egypt; khaled.elawdan@gmail.com (K.A.E.); shoroukelsaka.3@gmail.com (S.E.); salwa3eg@yahoo.com (S.E.M.)

**Keywords:** breast cancer, amino acid analogs, acivicin, azaserine, glutamine synthetase

## Abstract

Cancer cells undergo metabolic rewiring to support rapid proliferation and survival in challenging environments. Glutamine is a preferred resource for cancer metabolism, as it provides both carbon and nitrogen for cellular biogenesis. Recent studies suggest the potential anticancer activity of amino acid analogs. Some of these analogs disrupt cellular nucleotide synthesis, thereby inhibiting the formation of DNA and RNA in cancer cells. In the present study, we investigated the anticancer properties of Acivicin and Azaserine in the breast cancer MCF-7 cell line, comparing their effects to those on the non-tumorigenic MCF-10 epithelial cell line in vitro. Interestingly, at lower concentrations, both Acivicin and Azaserine showed potent inhibition of MCF-7 cell proliferation, as assessed by the MTT assay, without detectable toxicity to normal cells. In contrast, Sorafenib (Nexavar), a commonly used drug for solid tumors, showed harmful effects on normal cells, as indicated by increased lactate dehydrogenase (LDH) production in treated cells. Furthermore, unlike Sorafenib, treatment with Acivicin and Azaserine significantly affected apoptotic signaling in treated cells, indicating the role of both amino acid analogs in activating programmed cell death (PCD), as assessed by the Annexin-V assay, DAPI staining, and the relative expression of tumor suppressor genes PTEN and P53. ELISA analysis of MCF-7 cells revealed that both Acivicin and Azaserine treatments promoted the production of anti-inflammatory cytokines, including IL-4 and IL-10, while significantly reducing the production of tumor necrosis factor alpha (TNF-α). Mechanistically, both Acivicin and Azaserine treatment led to a significant reduction in the expression of glutamine synthetase (GS) at both the RNA and protein levels, resulting in a decrease in intracellular glutamine concentrations over time. Additionally, both treatments showed comparable effects on Raf-1 gene expression and protein phosphorylation when compared with Sorafenib, a Raf-1 inhibitor. Moreover, docking studies confirmed the strong binding affinity between Acivicin, Azaserine, and glutamine synthetase, as evidenced by their docking scores and binding interactions with the enzyme crystal. Collectively, these findings provide evidence for the anticancer activity of the two amino acid analogs Acivicin and Azaserine as antagonists of glutamine synthetase, offering novel insights into potential therapeutic strategies for breast cancer.

## 1. Introduction

Breast cancer development involves a complex sequence of cellular and molecular changes, often beginning with genetic mutations that transform normal cells into abnormal ones [1]. Over time, these mutations can trigger uncontrolled cell division, leading to tumor formation and, in some cases, metastasis, where cancer cells spread to other parts of the body [2,3]. Breast tissue consists of various cell types, including epithelial cells, which line the ducts and lobules (glandular tissue); stromal cells, which provide structural support; and myoepithelial cells, which help contract the ducts and lobules during lactation. The development of breast cancer typically begins with genetic mutations in the DNA of cells. These mutations can occur for several reasons, including inherit mutations in specific genes, such as BRCA1 or BRCA2, which are responsible for DNA repair [4,5]. Mutations in these genes impair the body’s ability to correct DNA damage, leading to an accumulation of additional genetic changes and a higher risk of cancer. In many cases, breast cancer arises from mutations that occur during a person’s lifetime, often due to exposure to environmental factors like radiation, chemicals, or lifestyle choices or from random errors during cell division [6]. Once a tumor becomes malignant, it may invade nearby tissues through local invasion. Cancer cells can also enter the bloodstream or lymphatic system, allowing the cancer to spread to distant organs in a process known as metastasis. Common sites for metastatic breast cancer include the bones, liver, lungs, and brain [7].

Glutamine is the most abundant amino acid in the human body, constituting 20% of the total free amino acids in the blood [8]. Cells acquire glutamine from circulation or through de novo synthesis by the enzyme glutamine synthetase (GS), which converts glutamate and ammonia into glutamine using adenosine triphosphate (ATP). GS is highly expressed in the liver, kidney, skeletal muscle, and brain, playing various crucial roles. In the liver, glutamine synthetase (GS) detoxifies ammonia, while in the kidney, it regulates acid–base balance by controlling ammonium availability [9,10]. Skeletal muscle utilizes glutamine synthesized by GS for energy production, and in the brain, GS in astrocytes helps regulate glutamate levels to protect neurons from excitotoxicity [11,12]. The structure of GS is well understood, with human GS existing as a decamer composed of two pentameric rings, which are held together by non-covalent interactions and hydrogen bonds. Each decamer has 10 active sites, located at the interface between the two pentameric rings. GS contains a β-grasp domain, as well as a catalytic domain involved in the synthesis of glutamine.

Moreover, the number of d-amino acids, particularly d-leucine, significantly affected the anticancer activity of Galaxamide analogs against in vitro cancer cell lines. In particular, the analog containing four d-leucine residues exhibited the greatest anticancer potential. To identify a more potent anticancer agent, further studies reported the synthesis and extensive evaluation of six additional Galaxamide analogs, focusing on their structure–activity relationships. Generally, when phenylalanine replaced leucine in the Galaxamide structure, there was a noticeable impact on its anticancer activity, suggesting that phenylalanine might interact with the active site to enhance its efficacy against cancer cells [13,14]. The expression of asparagine synthetase (ASNS) in cancer cells is notably elevated to meet the cell’s increased demand for asparagine (ASP), which plays a key role in various biological processes associated with cell proliferation and cancer progression. This overexpression of ASNS has been linked to a mechanism of resistance to asparaginase-based therapies, which are used to target asparagine in cancer treatment. A recent study highlighted that N, N-dibenzylasparagine can inhibit the proliferation of colon cancer cells while sparing normal human colon mucosal epithelial cells. The compound effectively modulated ASNS expression and induced programmed cell death (apoptosis) in cancer cells, suggesting its potential as a therapeutic strategy in overcoming asparaginase resistance and selectively targeting cancer cells [15]. It is noteworthy that X-ray crystallography of proteins has led to increasing interest in chalcogen analogs of amino acids in the methionine and serine/cysteine series, as reviewed in the previous literature. These analogs differ in their physicochemical properties, such as electronegativity, atomic volume, and carbon–metal bond length, depending on the nature of their side chains and the presence of chalcogen atoms. Additionally, fully synthetic chalcogen amino acids have been developed as isosteric analogs of tryptophan and phenylalanine. These have been used for in vivo protein synthesis, allowing the incorporation of chalcogen analogs into proteins for X-ray crystallography and other applications [16,17]. The current study was conducted to investigate the potential anticancer properties of two amino acid analogs, Acivicin and Azaserine, using breast cancer cell lines. Additionally, the study aimed to explore the molecular interactions that may be involved in their activity through in vitro and bioinformatics analyses.

## 2. Results

### 2.1. Selective Regulation of Breast Cancer Cell Proliferation by Acivicin and Azaserine Treatment

First, we investigated the possible antiproliferative effects of both Acivicin and Azaserine on the MCF-7 cell line compared to their effects on normal cells, specifically the MCF-10 cell line. Additionally, we evaluated the anticancer properties of Sorafenib, a commonly used drug for treating solid tumors. We then compared its effectiveness to that of Acivicin and Azaserine in inhibiting cancer cell growth and potential cytotoxicity in normal cells. Accordingly, the MCF-7 and MCF-10 cell lines were seeded in a 96-well plate and treated with various concentrations of the indicated drugs (3.125–50 μM) for 24 h. Treatment with an equal volume of DMSO served as a control. As shown in Figure 1A, MCF-7 cell viability was significantly affected by Acivicin, Azaserine, and Sorafenib in a dose-dependent manner compared to DMSO-treated cells, as indicated by the mean absorbance values obtained using the MTT assay. Treatment with the indicated concentrations of Sorafenib resulted in toxic effects on the MCF-10 cell line, as evidenced by a marked reduction in cell viability. In contrast, treatment with equivalent concentrations of Acivicin and Azaserine had a minimal impact on MCF-10 cell viability compared to the DMSO control, as shown in Figure 1B. Furthermore, while MCF-10 cells exhibited significant morphological changes following treatment with 6.25 μM of Sorafenib, cells treated with Acivicin and Azaserine at the same concentration maintained a morphology similar to the control cells. In the case of MCF-7 cells, all drug treatments led to notable changes in cell morphology, as observed under an inverted microscope (Figure 1C). Collectively, these results suggest that Acivicin and Azaserine are able to modulate cancer cells at low concentrations without causing detectable cytotoxicity in normal MCF-10 cells, in contrast to Sorafenib, which exhibits a toxic effect on normal cells at the same concentration.

### 2.2. Acivicin and Azaserine Exhibit Significant Apoptotic Effects with Less Overall Cytotoxicity in MCF-7 Cells

To check whether the treatment with the indicated drugs could modulate programmed cell death (PCD) in MCF-7 cells treated with 6.25 μM, we monitored the cells with early and late apoptotic signaling, in addition to monitoring the dead cells with the Annexin V assay using flow cytometry. As presented in Figure 2A–C the results indicate that Sorafenib, Acivicin, and Azaserine exert differential effects on PCD and necrotic events, as assessed by Annexin V staining and LDH release assays. DMSO-treated cells (control) showed minimal apoptotic signaling and a low percentage of dead cells, suggesting that the dissolving agent had no significant effect on PCD. Sorafenib treatment resulted in significant cell death, with almost 50% of cells undergoing apoptosis or necrosis, as evidenced by Annexin V staining. This suggests that Sorafenib has potent cytotoxic effects on MCF-7 cells, possibly via triggering necrotic events. Acivicin induced early apoptotic signaling in over 40% of the treated cells. This indicates that Acivicin primarily promotes early apoptosis without causing as much necrosis or late-stage apoptosis. Azaserine caused a shift toward late-stage apoptosis in about 25% of the treated cells, with a comparable percentage of dead cells (around 15%) to Acivicin. This suggests that Azaserine may act by inducing late apoptosis, but with less potency in terms of overall cell death. The LDH release assay confirmed that Sorafenib treatment led to a significant increase in LDH production, which is indicative of substantial cell membrane damage and cell death. The data suggest that Sorafenib’s cytotoxicity is likely due to necrosis effects. Acivicin and Azaserine treatments showed only slight increases in LDH production relative to control cells, suggesting these treatments are less effective at inducing cell membrane damage and are likely causing less severe cytotoxicity compared to Sorafenib (Figure 2C and Table 1). These findings suggest that while Sorafenib might be the most effective treatment in triggering cell death in MCF-7 cells, Acivicin and Azaserine also exhibit significant apoptotic effects, although through different mechanisms and with less overall cytotoxicity.

### 2.3. Acivicin and Azaserine Treatment Stimulates PCD via Restoring PTEN and P53 Gene Expression in Treated MCF-7 Cells

To further investigate the impact of Acivicin and Azaserine on cell survival in comparison to Sorafenib, MCF-7 cells were plated in a 24-well plate and incubated overnight. The cells were then treated with 6.25 μM of each drug and incubated for two time points, 12 h and 24 h. Cell survival was assessed using DAPI, a fluorescent dye that stains DNA. Interestingly, treatment with Acivicin resulted in fewer surviving cells at 12 h compared to Azaserine, which exhibited a similar reduction in cell survival to Acivicin at 24 h. Notably, Sorafenib treatment showed minimal effect on cell survival at 12 h, but led to a significant reduction in survival at 24 h (Figure 3A). Furthermore, the relative expression of the tumor suppressor genes PTEN and P53 was significantly increased, showing up to a three-fold increase in cells treated with Acivicin or Azaserine compared to those treated with Sorafenib or DMSO (Figure 3B,C). These results further support the role of Acivicin and Azaserine in promoting PCD through apoptotic signaling, as indicated by the elevated levels of PTEN and P53. In contrast, the data suggest a potential necrotic effect of Sorafenib on both cancer and normal cells.

### 2.4. Acivicin and Azaserine Treatment Positively Regulate the Production of Anti-Inflammatory Cytokines in MCF-7 Cells

To evaluate the effects of Acivicin and Azaserine on inflammatory cytokine production in treated MCF-7 cells, levels of IL-4, IL-6, and IL-10 were measured using an ELISA assay in a time-course experiment. Additionally, TNF-α, a marker of necrosis, was also monitored. All drug treatments resulted in a time-dependent increase in IL-6 production in treated cells, particularly following Acivicin and Azaserine treatment, when compared to control-treated cells (Figure 4A). Treatment with Sorafenib led to a time-dependent increase in TNF-α, reaching 700 pm/mL at 72 h, whereas Acivicin and Azaserine treatments showed TNF-α levels comparable to the control at the same time point (Figure 3B). Moreover, IL-4 and IL-10 levels increased significantly upon treatment with Acivicin and Azaserine in a time-dependent manner, reaching over 350 pm/mL for both cytokines at 72 h (Figure 4C,D). These results confirm that Acivicin and Azaserine contribute to reducing inflammatory events in treated MCF-7 cells by enhancing anti-inflammatory cytokines IL-4 and IL-10 and reducing TNF-α production. The rising levels of TNF-α in response to Sorafenib treatment further support the hypothesis, suggesting an overlap between Sorafenib’s effects and necrotic events in cancer cells.

### 2.5. Acivicin and Azaserine-Regulated GS as a New Insights Mechanism of Their Anticancer Properties

To assess the potential regulation of *Raf-1* and *glutamine synthetase* gene expression by Acivicin and Azaserine treatment in MCF-7 cells, their expression profiles were quantified using qRT-PCR and immunoblotting analyses. Interestingly, the relative gene expression of *Raf-1* was significantly reduced in cells pretreated with 6.25 μM of either Acivicin or Azaserine, as well as in cells pretreated with the same concentration of Sorafenib (Figure 5A). Similarly, the relative gene expression of *glutamine synthetase* was markedly decreased, showing more than 80% inhibition in MCF-7 cells pretreated with either Acivicin or Azaserine compared to cells pretreated with Sorafenib or control-treated cells (Figure 5B). Furthermore, the protein expression of phosphorylated Raf-1 (phospho-Raf-1) was significantly reduced in cells pretreated with Acivicin or Azaserine, as well as in cells pretreated with Sorafenib, in comparison to control-treated cells (Figure 5C). Notably, the kinetic protein expression of GS was only noticeably depleted in cells pretreated with either Acivicin or Azaserine, suggesting that both drugs specifically inhibit the GS expression profile in MCF-7 cells (Figure 5C). Furthermore, we monitored the intracellular concentration of glutamine in MCF-7 cells over time following treatment with the indicated drugs, using the Glutamine-WST assay kit. Our findings revealed an increasing level of intracellular glutamine in both the control-treated MCF-7 cells and the cells pretreated with Sorafenib (Figure 5D). Interestingly, the intracellular glutamine concentration significantly decreased in a time-dependent manner in response to Acivicin and Azaserine treatment (Figure 5D). These results suggest that both Acivicin and Azaserine effectively modulate total glutamine concentration in MCF-7 cells through direct interaction with GS. Collectively, these findings provide new insights into the potential mechanisms underlying the anticancer properties of Acivicin and Azaserine in breast cancer cells.

### 2.6. Acivicin and Azaserine Show High Docking Scores and Strong Binding Affinity for GS

The interaction between the crystal structure of GS and the ligands Acivicin, Azaserine, and Sorafenib was explored in greater detail and compared with the interaction of GS with its standard inhibitor, Methionine sulfoximine (MS), using docking analysis via the SwissDock online tool. This analysis provided insights into the docking scores, binding affinities, and free energy between the ligands and the protein domains. As shown in Figure 6, several binding sites with high docking scores and low energy requirements were identified for Acivicin and Azaserine compared to the standard inhibitor, MS. Specifically, more than 38 binding sites were found between Acivicin and GS, while nearly 52 binding affinities were detected between Azaserine and GS (Table 2). These results suggest that Acivicin and Azaserine could disrupt GS function through direct interactions, as evidenced by the high number of binding affinities observed in their crystal structure.

## 3. Discussion

Cancer cells need a lot of glutamine to sustain their random and rapid growth; accordingly, the crucial impact of GS and glutamine on tumor metabolism has been broadly stated [18]. Here, we provide evidence for the potential anticancer activity of two amino acid analog antagonists of GS, Acivicin and Azaserine. These amino acid analogs interrupt cellular nucleotide synthesis and thereby inhibit the formation of DNA and/or RNA molecules in the cancer cell. Sorafenib appears to be the most effective in inducing cell death in MCF-7 cells, likely due to its potent cytotoxic effects, which could involve a combination of apoptosis and necrosis pathways. MCF-7 cells are a widely used breast cancer cell line and can secrete various cytokines depending on their microenvironment. They are known to exhibit some immune-modulating properties, especially in the context of tumor progression [19]. Here, the treatment with Acivicin or Azaserine altered the secretion profiles of pro-inflammatory cytokines (IL-6 and TNF-α), which are induced under stress conditions, while also increasing the production of anti-inflammatory cytokines (IL-4 and IL-10) as part of a compensatory mechanism to counteract excessive inflammation during tumor development. This could involve activation of the nuclear factor kappa B (NF-κB) pathway or the JAK-STAT signaling cascade, which are both involved in immune responses and cytokine regulation [20].

Acivicin and Azaserine could alter the levels of inflammatory cytokines by shifting the balance of metabolic intermediates, leading to a compensatory inflammatory response. These compounds might reduce the expression of pro-inflammatory cytokines (like TNF-α and IL-6) while potentially enhancing the production of anti-inflammatory cytokines (like IL-4 or IL-10) as the cell attempts to adapt to stress. The crosslink between Acivicin and Azaserine may involve a balance between pro-inflammatory and anti-inflammatory cytokines, influencing the tumor microenvironment in a way that might either promote immune tolerance or help the tumor evade immune detection. The shift toward anti-inflammatory cytokine production could play a role in increasing immune system recognition of tumor cells, potentially making it more sufficient for the immune system to mount an effective anti-tumor response [21]. On the other hand, certain anti-inflammatory cytokines could suppress tumor growth by promoting a less permissive tumor environment [22].

Recent evidence has demonstrated that resveratrol, when included in the diet, can reduce the formation of atypical acinar cell foci (AACF) in the exocrine pancreas of rats treated with Azaserine, ultimately decreasing tumor burden. This suggests that resveratrol may play a role in mitigating the development of pancreatic cancer. Known for its potential anticancer properties, resveratrol is found in grapes and other foods [23]. Azaserine, a compound with a biologically unique α-diazoester functional group, acts as a DNA carboxymethylation agent [24]. It has been shown to induce DNA damage and telomere dysfunction in acute myeloid leukemia (AML) cell lines, as evidenced by the formation of 53-BP1 foci and their co-localization with telomere signals. This telomere dysfunction is associated with decreased expression of telomerase reverse transcriptase (TERT), shortened telomeres, and increased apoptosis in Azaserine-treated cells. However, Azaserine does not appear to affect the methylation status of subtelomere regions. In primary leukemic cells from AML patients, Azaserine similarly down-regulated TERT expression. Notably, overexpression of TERT in AML cells alleviated the DNA damage, telomere dysfunction, and apoptosis induced by Azaserine. These findings indicate that Azaserine causes down-regulation of TERT and telomere dysfunction, contributing to its anti-tumor effects [25]. Acivicin is a natural compound with a wide range of biological activities. Although its potential for cancer treatment was investigated decades ago, clinical use was discontinued due to excessive toxicity. The underlying causes of its beneficial and harmful effects have remained unclear, and limited information is available regarding Acivicin’s specific molecular targets. To gain a deeper understanding of its target profile, we synthesized functionalized derivatives and employed activity-based proteomic profiling (ABPP) in intact cancer cells. Quantitative mass spectrometry (MS) revealed a strong preference for targeting specific aldehyde dehydrogenases. Further validation studies confirmed that Acivicin inhibits the activity of ALDH4A1 by binding to its catalytic site. Consistent with these findings, siRNA-mediated downregulation of ALDH4A1 and significantly impaired cell growth, providing a potential explanation for Acivicin’s cytotoxic effects [26]. Our findings likely revealed that Acivicin induced early apoptosis, but its effect on cell death is less pronounced than that of Sorafenib, showing a moderate impact on cells without causing significant membrane damage. Azaserine treatment resulted in late-stage apoptosis, but similarly, its overall effect on cell death is lower than Sorafenib’s, and its LDH release profile suggests less-severe membrane damage. In contrast, Sorafenib demonstrated a clear and strong cytotoxic effect, inducing the late stages of cell death, and significantly increased LDH production, indicating more pronounced necrotic responses. Overall, Acivicin and Azaserine exhibited milder effects, inducing early or late apoptosis with less membrane damage and lower overall cell death compared to Sorafenib.

Notably, GS is an enzyme responsible for converting glutamate and ammonia to glutamine [12,27]. It is a target for certain drugs, including Acivicin and Azaserine, and could affect cellular signaling pathways that influence the production of cytokines, including pro-inflammatory cytokines such as TNF-α. Therefore, glutamine depletion arises in various types of cancer, particularly in poorly vascularized tumors. This makes GS, the only enzyme responsible for the de novo synthesizing of glutamine, a vital enzyme in tumor development [18]. In cancer, GS displays a pro-tumoral role by synthesizing glutamine, supporting nucleotide synthesis [28]. Additionally, GS supplies glutamine to cancer cells, facilitating cancer cells to maintain adequate glutamine levels for glutamine catabolism. Therefore, if expression is increased in the tumor microenvironment (TME) and glutamine catabolism, the conflicting response of glutamine synthesis by GS is known for promoting cancer cell proliferation via encouraging biosynthesis of critical molecules and energy production. Both glutamine anabolism and catabolism play an essential role in cancer metabolism depending on the complex nature and microenvironment of cancers [28]. Glutaminolysis is a crucial process in cancer metabolism, involving the breakdown of glutamine by the enzyme glutaminase (GLS) to produce energy [29,30].

Interestingly, through bioinformatics analysis, the high docking score obtained from the interaction between Acivicin or Azaserine and GS means that Acivicin and Azaserine fit well into the active site of the GS enzyme. A high docking score suggests that the interaction between the ligand (Acivicin or Azaserine) and the enzyme is energetically favorable. The binding affinity refers to the strength of the interaction between the ligand (Acivicin or Azaserine) and the enzyme (GS). The high binding affinity means that the compound binds tightly to the enzyme, which could imply effectiveness in inhibiting its activity.

## 4. Material and Methods

### 4.1. Cell Lines

The human breast adenocarcinoma MCF-7 cell line and the non-tumorigenic MCF-10 epithelial cell line were kindly provided by VACSERA in Giza, Egypt. These cells were cultured in Roswell Park Memorial Institute (RPMI) 1640 medium, supplemented with 25 mM HEPES, 4 mM l-glutamine, 4 mM sodium pyruvate, and 10% heat-inactivated bovine serum albumin (BSA). Cell propagation was performed in a CO_2_ incubator at 37 °C with 95% relative humidity [31]. The cultured cells were observed and imaged using inverted microscopy (Olympus CKX53, Tokyo, Japan), with a Zeiss A-Plan 10× objective lens (White Plains, NY, USA).

### 4.2. Preparation of Drugs

Sorafenib (Nexavar, molecular weight 464.82, SML2653, Sigma-Aldrich, Rockville, MD, USA), Acivicin (molecular weight 178.6, A2295, Sigma-Aldrich, Rockville, MD, USA), and Azaserine (molecular weight 173.13, A4142, Sigma-Aldrich, Rockville, MD, USA) were prepared in Dimethyl sulfoxide (DMSO) at concentrations of 50 μM, 25 μM, 12.5 μM, 6.25 μM, and 3.125 μM. The resulting dilutions were transferred into clean tubes and stored at 4 °C until use.

### 4.3. Proliferation Assay Cytotoxicity

MCF-7 and MCF-10 cells were seeded in triplicate in 96-well plates at a density of 10,000 cells per well (10 × 10^3^ cells) and incubated overnight at 37 °C under humidified conditions. The cells were then treated with various concentrations of each drug, ranging from 50 μM to 3.125 μM, and incubated for another 24 h. Cell viability and the cytotoxic concentration 50% (CC50) were determined using the MTT colorimetric assay (Sigma-Aldrich, Rockville, MD, USA). Briefly, after the treatment period, the culture medium was discarded, and the cells were washed with phosphate-buffered saline (PBS). Next, 100 μL of complete RPMI medium was added to each well, followed by the addition of 10 μL of MTT solution to each well. The plate was incubated for 2 h at 37 °C. After incubation, 100 μL of SDS-HCl solution was added to each well, and the plate was incubated for an additional 4 h at 37 °C. Cell viability was assessed by measuring the optical density at 570 nm, based on the conversion of water-soluble MTT to an insoluble formazan, which was then solubilized with SDS-HCl solution. The amount of formazan formed is proportional to the number of viable cells in each well, allowing determination of the cell viability rate and the CC50 [15]. Cell morphology of treated cells was observed using an inverted microscope. In brief, cells were seeded in duplicate wells in a 6-well plate at a density of 1 × 10⁴ cells per well. The cells were treated with 6.25 μM of each agent and incubated for 24 h under the same conditions described previously. The survival rate of the cells was determined using a hemocytometer. First, the old medium was discarded, and the cells were washed twice with phosphate-buffered saline (PBS). Trypsin was then added to detach the cells, and the cells were incubated at 37 °C for 3 min. After trypsinization, 8 mL of complete RPMI medium was added to neutralize the trypsin. The cell count was determined by observing 1 μL of the cell suspension under an inverted microscope [32].

### 4.4. Annexin-V Assay

Apoptosis analysis was conducted using an Annexin-V (FITC)/propidium iodide (PI) assay kit (BD Biosciences, Beijing, China) following the manufacturer’s protocol. In brief, 1 × 10^5^ MCF-7 cells were cultured in a 25 mL flask and incubated overnight under standard culture conditions. The cells were then treated with 20 μM of Sorafenib, Acivicin, or Azaserine for 24 h. After treatment, the cells were harvested, washed twice with PBS, and resuspended in the provided Binding Buffer. To detect apoptosis, 100 μL of the cell suspension was incubated with 5 μL of Annexin-V conjugated to fluorescein isothiocyanate (FITC) and 5 μL of propidium iodide (PI) for 15 min at room temperature in the dark. After incubation, 500 μL of Binding Buffer was added, and the samples were immediately analyzed by flow cytometry within one hour to assess early and late apoptosis [33,34].

### 4.5. Cell Staining and Fluorescent Assay

MCF-7 cells were seeded onto coverslips in 24-well plates at a density of 2 × 10^4^ cells per well and allowed to incubate overnight. The cells were then treated with 6.25 μM of each drug and incubated for either 12 or 24 h, with DMSO-treated cells serving as the control. After the respective incubation periods, cells were fixed in 4% paraformaldehyde for 25 min at room temperature (RT), followed by permeabilization with cold methanol for 10 s. The cells were then washed three times with PBS and stained with the fluorescent DNA dye DAPI (1 μg/mL) for 30 min. After additional washing, fluorescent images were captured using a laser scanning confocal fluorescence microscope (Olympus FluoView FV10i, Tokyo, Japan) with a 10× objective [31].

### 4.6. Glutamine Assay

MCF-7 cells were seeded in 6-well plates at a density of 1 × 10^5^ cells/well and incubated overnight in a CO_2_ incubator at 37 °C with 95% relative humidity. The next day, the cells were treated with 6.25 μM of each indicated drug (Sorafenib, Acivicin, or Azaserine). Cells treated with the same volume of DMSO served as the control group. The cells were incubated for different time points, including 0, 12, 24, 36, 48, and 60 h. At each time point, cells were detached using trypsin, collected in suspension in 1.5 mL microtubes, and centrifuged at 300× *g* for 5 min. The supernatant was carefully removed. The cell pellets were resuspended in 500 μL PBS and gently pipetted to wash the cells, followed by centrifugation at 300× *g* for 5 min and removal of the supernatant. To lyse the cells, 400 μL of lysis buffer was added to each tube and cells were pipetted carefully to ensure complete lysis. The tubes were centrifuged at 12,000× *g* for 5 min, and 350 μL of the supernatant was transferred to a filtration tube. The filtrate was centrifuged for 10 min at 12,000× *g*. To prepare the standard curve for glutamine concentration, 50 μL of the glutamine standard stock solution (10 pm/mL) was mixed with 950 μL of deionized and distilled water (ddH_2_O) in a microtube to prepare a 0.5 pm/mL glutamine standard solution. Serial dilutions were performed using ddH_2_O to generate the following standard concentrations: 0.5, 0.25, 0.125, 0.0625, 0.0313, 0.0157, 0.00785, and 0 pm/mL. The filtered samples and prepared standard solutions were loaded onto an ELISA plate reader in triplicate. To each well, 20 μL of glutaminase solution was added, followed by gentle mixing. Then, 20 μL of reaction buffer was added to each well, and the plate was incubated at 37 °C for 15 min. After incubation, 60 μL of working solution was added to each well, and the plate was incubated at 37 °C for an additional 30 min. Finally, absorbance was measured at 450 nm using a microplate reader. The final glutamine concentration in each sample was calculated using the calibration curve derived from the glutamine standards.

### 4.7. Quantitative RT-PCR Procedure

Total RNA was extracted from treated MCF-7 cells using the TriZol reagent (Invitrogen, Carlsbad, CA, USA), followed by chloroform and isopropanol precipitation. The RNA was then purified using an RNA purification kit (Invitrogen, USA) according to the manufacturer’s instructions. Purified RNA was used to synthesize complementary DNA (cDNA) with the QuantiTech Reverse Transcription Kit (Qiagen, Carlsbad, CA, USA), following the manufacturer’s protocol [35]. The relative expression levels of Raf-1 and glutamine synthetase were assessed in the treated cells and normalized to the expression levels in untreated cells, using specific primers (Table 3). Gene amplification was performed using the QuantiTect SYBR Green PCR Kit (Qiagen, Carlsbad, CA, USA), with GAPDH as a housekeeping control. The PCR conditions included an initial denaturation at 94 °C for 5 min, followed by 40 cycles of 94 °C for 30 s, 60 °C for 30 s, and 72 °C for 45 s [36,37].

### 4.8. Western Blot

Protein expression levels of c-Raf and glutamine synthase in treated MCF-7 cells were analyzed using immunoblotting. Total protein was extracted from the cells with RIPA lysis and extraction buffer (Thermo Fisher, Rockford, IL, USA). The proteins were then denatured by adding a loading buffer containing sodium dodecyl sulfate (SDS) and boiled at 95–100 °C for 5 min. Equal amounts of denatured protein (100 ng) were loaded onto a 15% sodium dodecyl sulfate/polyacrylamide gel electrophoresis (SDS-PAGE) gel, and electrophoresis was performed for 3 h at 100 V using the Bio-Rad Mini-Protean II system. Following electrophoresis, the proteins were transferred to nitrocellulose membranes (Sigma-Aldrich, Millipore, MA, USA) using the Bio-Rad Mini Trans-Blot system at 250 mA for 3 h at 4 °C. The membranes were blocked with 30 mL of 5% non-fat dry milk in tris-buffered saline with Tween-20 (0.05 M Tris-HCl, 0.15 M NaCl, 0.1% Tween-20, pH 7.5) and incubated overnight at 4 °C with either rabbit monoclonal anti-pho-Raf-1 (1:500; Ab 173539, Abcam, USA) or rabbit polyclonal anti-GS (1:500; Ab 93439, Abcam, Rockville, MD, USA). After washing with Western Breeze solution 16x (Invitrogen, USA), the membranes were incubated for 2 h with mouse monoclonal anti-β-actin (Sigma, Hamburg, Germany). The membranes were then washed and incubated for 2 h at room temperature with alkaline phosphatase-conjugated secondary antibodies (anti-mouse or anti-rabbit) (Invitrogen, Carlsbad, CA, USA). Finally, the protein bands were visualized using a chromogenic substrate for alkaline phosphatase (AP substrate, Western Breeze, Invitrogen, Carlsbad, CA, USA) [38,39].

### 4.9. Enzyme-Linked Immunosorbent Assay (ELISA)

The release of interleukin-4 (IL-4), IL-6, IL-10, and TNF-α was quantified using human ELISA kits Abcam ab46063, ab46027, ab46034, and ab46087 (Rockville, MD, USA), respectively. As described previously, MCF-7 cells were seeded in a 96-well plate and treated with 20 μM of each drug. The cells were incubated for various time points (0, 6, 12, 24, 48, and 72 h). At the end of each incubation period, 100 μL of cell lysis buffer (Invitrogen, USA) was added to each well, mixed gently, and transferred to the ELISA plate. To each well, 50 μL of control solution and 50 μL of 1X biotinylated antibody were added and incubated at room temperature for 2 h. Following incubation, 100 μL of 1X streptavidin-HRP reagent was added to each well and incubated in the dark for 30 min. Afterward, 100 μL of TMB chromogen substrate was added to each well, and the reaction was allowed to develop for 15 min at room temperature, protected from light. The reaction was stopped by adding 100 μL of stop reagent to each well. Absorbance was measured at 450 nm to determine the levels of cytokines [38,40,41].

### 4.10. Molecular Docking Study

Molecular docking analysis was carried out using the SwissDock online tool with the systematic structures of Sorafenib, Acivicin, and Azaserine. The crystal structure of glutamine synthetase (PDB code: 20JW) was obtained from the RCSB Protein Data Bank (https://www.rcsb.org/structure/2ojw, accessed on 1st November 2024). Each ligand was docked individually into the binding site of glutamine synthetase using the default parameters, and multiple docking poses were generated for further visual inspection. The docking scores, binding affinities, and free energy values for each pose were automatically computed by SwissDock.

### 4.11. Data Analysis

The quantification analysis of gene expression was calculated by normalizing the cycle threshold (Ct) value for each target gene to GAPDH and by normalizing between samples and the control group. Relative gene expression levels were determined using the delta-delta Ct method, with the results presented as fold changes.

## 5. Conclusions

Recent studies have suggested the potential antioxidant and anticancer activities of amino acid analogs. Some of these analogs disrupt cellular nucleotide synthesis, thereby inhibiting the formation of DNA and RNA in cancer cells. In this context, the current study provides the first evidence for the anticancer properties of Acivicin and Azaserine, two amino acid analogs, in MCF-7 cell lines. Importantly, neither of these analogs exhibited detectable toxic effects on normal MCF-10 cells. Treatment with very low concentrations of Acivicin or Azaserine effectively stimulated apoptotic signaling in treated cells by reducing Raf-1 protein activation and decreasing the expression levels of glutamine synthetase (GS). The targeting of GS by Acivicin and Azaserine was further confirmed through the high docking scores and binding affinities observed in the docking analysis. Moreover, it is plausible that the metabolic stress induced by these compounds leads to a shift in the production of anti-inflammatory cytokines. This shift may offer new insights into therapeutic approaches that target both metabolism and immune modulation in breast cancer.

## Figures and Tables

**Figure 1 ijms-26-00078-f001:**
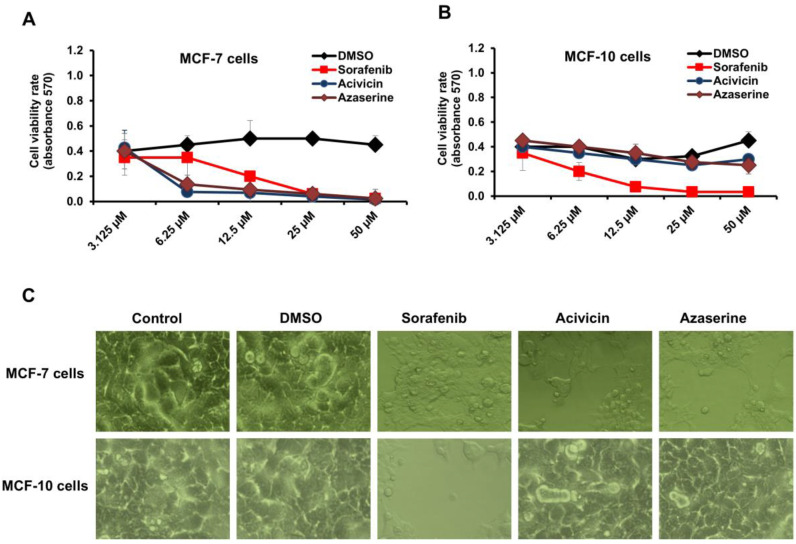
Cell viability and cytotoxicity of Acivicin and Azaserine in breast cancer and normal breast cell lines. (**A**) Cell viability rate of the indicated amino acid analog treatment on MCF-7 cells in response to different concentrations of Acivicin and Azaserine compared to the same concentrations of Sorafenib and DMSO treatment. Error bars indicate the standard deviation (SD) of four different replicates. (**B**) Cell viability rate of the indicated amino acid analog treatment of MCF-10 cells in response to different concentrations of Acivicin and Azaserine compared to the same concentrations of Sorafenib and DMSO treatment. (**C**) MCF-7 and MCF-10 cell morphology presented by inverted microscope (10× magnification) upon 24 h treatment with 6.25 μM of each indicated analog of amino acids compared with the treatment with the same concentration of Sorafenib and DMSO treatment.

**Figure 2 ijms-26-00078-f002:**
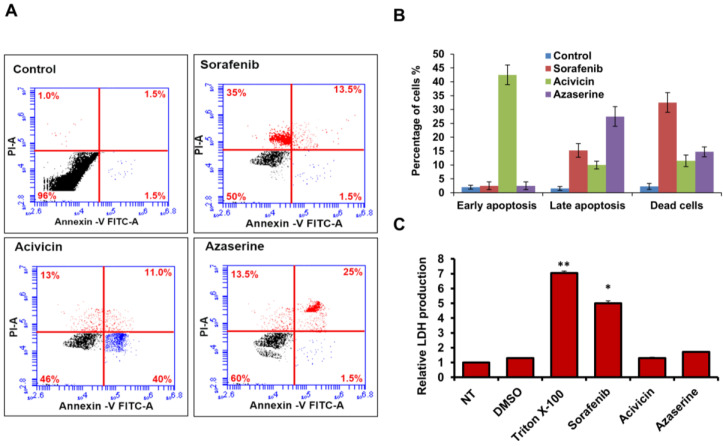
Apoptotic response and LDH production in response to amino acid analog treatment. (**A**) MCF-7 cells were treated with 6.25 μM of each indicated amino acid analog and then cells were stained with (Annexin V+/Propidium Iodide (PI)). The early and late apoptotic singling and dead cells were monitored using flow cytometry. The cells in early apoptosis were identified in the lower right quadrant, marked by blue dots, while cells in late apoptosis were located in the upper right quadrant, indicated by red dots. Dead cells were found in the upper left quadrant, also marked by red dots. (**B**) The percentage of treated cells with positive signals for early or late apoptosis and the percentage of dead cells indicated by flow cytometric assay. (**C**) Relative LDH production of treated cells with the indicated amino acid analogs in comparison with control-treated cells, Triton X-100, and DMSO-treated cells. Error bars indicate the SD of three independent experiments. Student’s two-tailed *t*-test used for statistical analysis; (*) indicates *p*-values ≤ 0.05, (**) indicates *p* ≤ 0.01.

**Figure 3 ijms-26-00078-f003:**
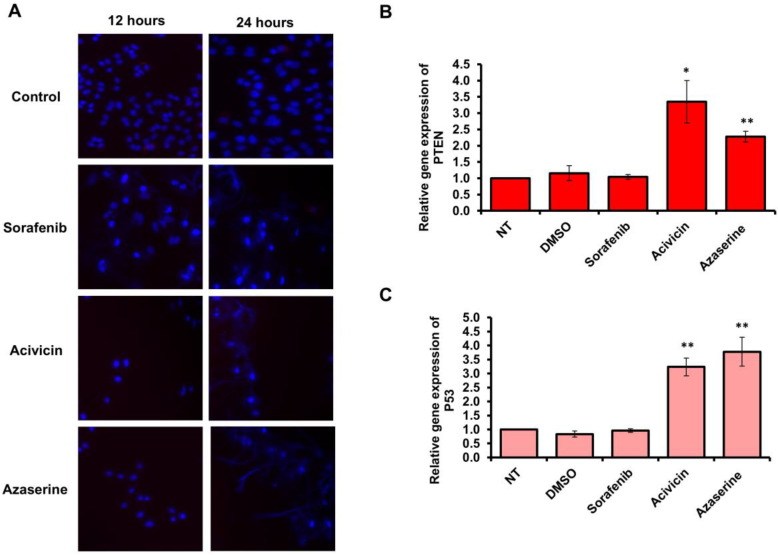
Cell survival and expression of tumor suppresser genes. (**A**) Fluorescent microscope images (40× magnification) of treated MCF-7 cells showing DNA staining with DAPI, used as an indicator of live cells, at 12 and 24 h after drug treatment. (**B**,**C**) Quantitative analysis of PTEN and p53 gene expression levels in treated cells was performed using fold changes from qRT-PCR. Error bars represent the SD. Statistical significance of the cycle threshold (Ct) values was assessed using a two-tailed Student’s *t*-test, with (*) indicates *p*-values ≤ 0.05 and (**) indicating a *p*-value of <0.01, denoting high significance. Data are representative of three independent experiments.

**Figure 4 ijms-26-00078-f004:**
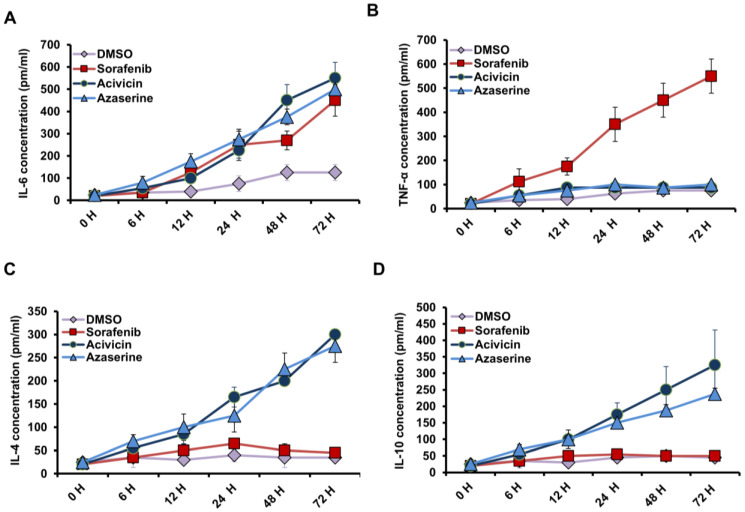
Levels of produced cytokines IL-4, IL6, IL10, and TNF-α in treated MCF-7 cells. (**A**,**B**) The concentrations of IL-6 and TNF-α (pm/mL) produced over time in the fluid media of MCF-7 cells that were pretreated with 6.25 μM Acivicin or Azaserine compared to cells treated with the same concentration of Sorafenib and control-treated cells. Error bars represent the SD of four different replicates. (**C**,**D**) The concentrations of IL-4 and IL-10 (pm/mL) produced over time in the fluid media of MCF-7 cells that were pretreated with 6.25 μM Acivicin or Azaserine, compared to cells treated with the same concentration of Sorafenib and control-treated cells. Error bars represent the SD of four different replicates.

**Figure 5 ijms-26-00078-f005:**
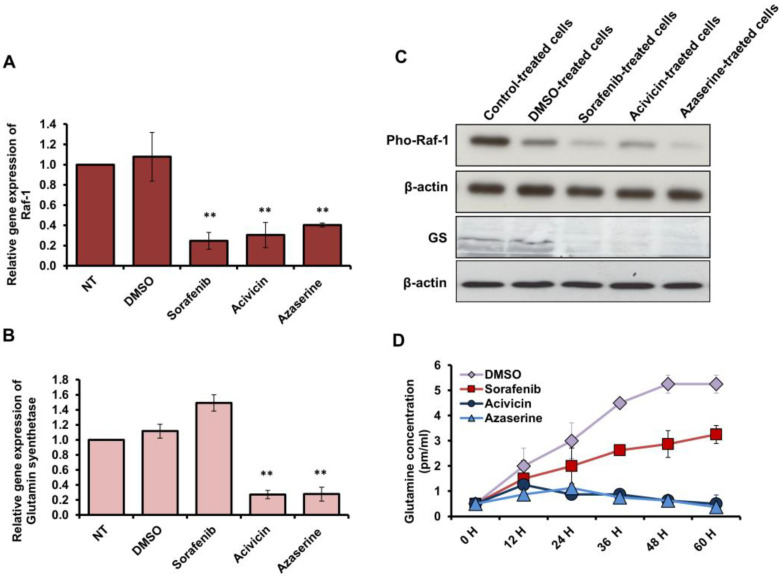
Quantification of GS and Raf-1 expression profiles in treated MCF-7 cells. (**A**,**B**) Steady-state mRNA levels of Raf-1 and GS, quantified as fold changes, were measured in MCF-7 cells treated with 6.25 μM of various drugs, compared to DMSO and control-treated cells. Error bars represent the SD from two independent experiments. A Student’s two-tailed *t*-test was used for significance analysis of cycle threshold (Ct) values, with (**) indicating *p* < 0.01, considered highly significant. (**C**) Protein levels of phospho-Raf-1 and GS were quantified in treated MCF-7 cells through immunoblotting analysis and compared to DMSO and control-treated cells. β-actin served as the internal control. (**D**) The total concentrations of glutamine over time in MCF-7 cells treated with 6.25 μM Acivicin or Azaserine were measured and compared to the respective concentrations in cells treated with Sorafenib, DMSO, and control treatments. Error bars represent the SD from four independent replicates. Data are representative of three independent experiments.

**Figure 6 ijms-26-00078-f006:**
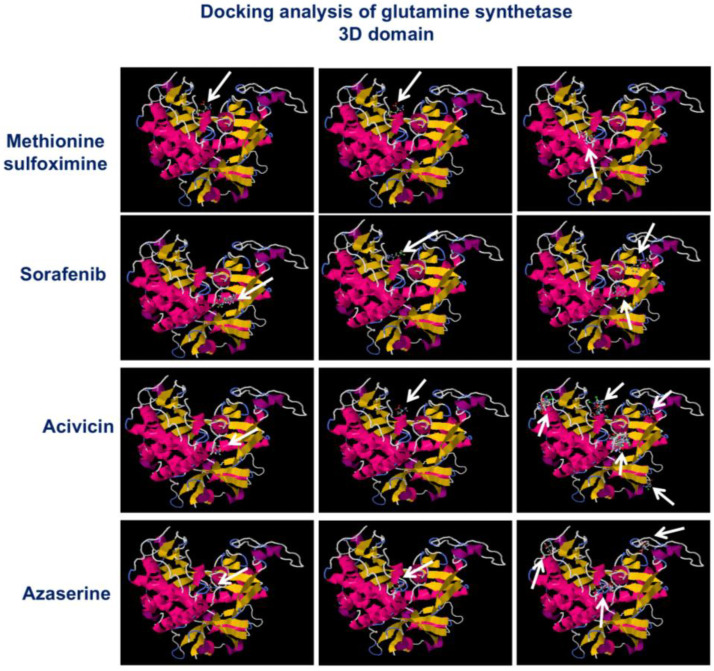
Docking analysis. The figure illustrates a typical output from SwissDock, showing the docking positions of GS with MS, Sorafenib, Acivicin, and Azaserine. Visual analysis was conducted using the ViewDock plugin of UCSF Chimera. The predicted BM of GS (represented by magenta sticks) is overlaid with the X-ray BM (shown as ball and sticks). As indicated by the white arrows, this particular predicted BM demonstrates the most favorable energy. Binding affinities of Acivicin, Azaserine, and GS crystal structures obtained through docking analysis using SwissDock software (http://old.swissdock.ch/docking/view/swissdockd_Cq1lhe_K7JARBW1YPTXICSDE1KQ accessed on 1 November 2024) indicates the potential binding affinities of the ligands MS to the GS protein structure compared with the original binding affinity of Sorafenib and MS, the standard GS inhibitor.

**Table 1 ijms-26-00078-t001:** LDH production levels in MCF-7 cells pretreated with 6.25 μM of each indicated drug.

	NT	DMSO	Triton X-100	Sorafenib	Acivicin	Azaserine
Mean absorbance	0.05	0.07	0.37 **	0.25 *	0.07	0.09
SD	0.02	0.02	0.13	0.17	0.06	0.04
Relative LDH production	1.00	1.33	7.05	5	1.31	1.71
*p* values		0.33	0.002	0.04	0.99	0.15

NT: nontreated cell, SD: standard deviation, *: Indicates high significant *p* values ≤ 0.05, **: Indicates high significant *p* values ≤ 0.01.

**Table 2 ijms-26-00078-t002:** Binding affinities and required energy in docking analysis of Sorafenib, Acivicin, and Azaserine with GS protein.

Glutamine Synthetase Domain	Binding Affinity	Full Fitness (kcal/mol) From–To	Estimated ΔG (kcal/mol) From–To
MS	35	−2168	(−7.04)
		−2147	(−5.66)
Sorafenib	31	−2166.96	(−8.27)
		−2146.21	(−8.44)
Acivicin	38	−2162.96	(−6.27)
		−2135.21	(−6.29)
Azaserine	52	−2154.96	(−6.21)
		−2130.21	(−6.11)

**Table 3 ijms-26-00078-t003:** Oligonucleotide sequences used for mRNA quantification of indicated genes.

Description	Primer Sequences 5′–3′
Raf-1-sense	TTTCCTGGATCATGTTCCCCT
Raf-1-antisense	ACTTTGGTGCTACAGTGCTCA
GS1-sense	ACTGGGTTCCACAAAACGTC
GS1-antisense	GATGGCTTCTGTCACTGCAA
PTEN-sense	TTCCATCCTGCAGAAGAAGC
PTEN-antisense	CTACGGACATTTTCGCATCC
P53-sense	GCGAGCACTGCCCAACAACA
P53-anisense	GGTCACCGTCTTGTTGTCCT
GAPDH-sense	TGGCATTGTGGAAGGGCTCA
GAPDH-antisense	TGGATGCAGGGATGATGTTCT

## Data Availability

Data that support the findings of this study will be made available from the corresponding author upon reasonable request.

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
