# Peer review of "Effective Targeting of Glutamine Synthetase with Amino Acid Analogs as a Novel Therapeutic Approach in Breast Cancer"

_ijms, 2024, doi:10.3390/ijms26010078_

Round 1

Reviewer 1 Report

Comments and Suggestions for Authors

Effective Targeting of Glutamine Synthetase by Amino Acid Analogs as a Novel Therapeutic Approach in Breast Cancer by Abdesttar S et al. and Khalil H, describes an interesting approach to understanding metabolic weaknesses dealing with amino acids needs of cancer, energy production and protein syntheses.

Of notice, there is growing interest and  different recent reports about this field of study in literature.

In vitro data showed that Acivicin and Azaserine effectively modulate total GS in MCF-7 cells through direct interaction with glutamine synthetase and also at molecular level, they report a blunted epression of the enzyme, and finally a decreased intracellular glutamine. Additionally, both treatments showed comparable effects on Raf-1 gene expression and protein phosphorylation when compared with Sorafenib (and this should be preferred to the use of Nexavar in this paper, in my opinion), a Raf-1 inhibitor.

Also, authors link those activities to some immune-modulation. They report altered secretion profiles of pro-inflammatory cytokines (IL-6 and TNF-α), but  also increased production of anti-inflammatory cytokines (IL-4 and IL-10) as part of a compensatory mechanism possibly to counteract excessive inflammation during tumor development. They suggest that this could involve activation of the nuclear factor kappa B (NF-κB) pathway or the JAK-STAT signaling cascade, which are both involved in immune responses and cytokine regulation. This point is worth of some disucssion: if those pro-inflamamtory cytokines (il-6, TNF-α) are blunted, where is the necessity to counteract excessive inflammation by IL-4 and IL-10?

I may agree that, as Abdesttar S et al. state, “ the shift toward anti-inflammatory cytokine production could play a role in increasing immune system recognition of tumor cells, potentially making it more sufficient for the immune system to mount an effective anti-tumor response [citation 21 in tet]. On the other hand, certain anti-inflammatory cytokines could suppress tumor growth by promoting a less permissive tumor environment [citation 22 in text]”, still the eventual relationship among glutamine synthase expression and the different interleukins would be worth of some discussion, at least as author’s advice.

Suggestions.

I suggest that authors would provide some information about uses and documented activities of those molecules, including eventual toxicities (g.e.: about toxicity of Azaserine: doi: 10.1111/apm.13498).

Few notes:

-in Fig.3A: IL-6 is or is not blunted?

-in page 4:

Nexavar treatment resulted in significant cell death, with almost 50% of cells undergoing apoptosis or necrosis, as evidenced by Annexin V staining. This suggests that Nexavar has potent cytotoxic effect

on MCF-7 cells, likely triggering both early and late stages of apoptosis, as well as necrosis. Acivicin induced early apoptotic signaling in over 40% of the treated cells. This indicates that Acivicin primarily promotes early apoptosis without causing as much necrosis or late-stage apoptosis. Azaserine caused a shift toward late-stage apoptosis in about 25% of the treated cells, with a comparable percentage of dead cells (around 15%) to Acivicin.

According to 4.3. Proliferation assay Cytotoxicity, if I understood correctly, cells were incubated for 48 plus 6 hours for MTT evaluation. Thus, after 48 h of incubations with the said molecules cytotoxicity was successful in 50% of Sorafenib treated cells, 40% of Azaserine and 25% of Acivicin treted cells. Is this understanding correct?

- in page 7:

Our findings revealed an increasing level of intracellular glutamine in both the control-treated MCF-7

cells and the cells pretreated with Nexavar (Figure 4D). Interestingly, the intracellular glutamine concentration increased (misprint  for: not increased ?) in a time-dependent manner in response to Acivicin and Azaserine treatment (Figure 4D). These results suggest that both Acivicin and Azaserine effectively modulate total GS in MCF-7 cells through direct interaction with glutamine synthetase.

Author Response

Reviewers Response

Reviewer 1

In vitro data showed that Acivicin and Azaserine effectively modulate total GS in MCF-7 cells through direct interaction with glutamine synthetase and also at molecular level, they report a blunted expression of the enzyme, and finally a decreased intracellular glutamine. Additionally, both treatments showed comparable effects on Raf-1 gene expression and protein phosphorylation when compared with Sorafenib (and this should be preferred to the use of Nexavar in this paper, in my opinion), a Raf-1 inhibitor.

We appreciate the reviewer's suggestion to use "Sorafenib" instead of "Nexavar" in the manuscript. In response, we have updated the manuscript accordingly.  

Also, authors link those activities to some immune-modulation. They report altered secretion profiles of pro-inflammatory cytokines (IL-6 and TNF-α), but also increased production of anti-inflammatory cytokines (IL-4 and IL-10) as part of a compensatory mechanism possibly to counteract excessive inflammation during tumor development. They suggest that this could involve activation of the nuclear factor kappa B (NF-κB) pathway or the JAK-STAT signaling cascade, which are both involved in immune responses and cytokine regulation. This point is worth of some disucssion: if those pro-inflamamtory cytokines (il-6, TNF-α) are blunted, where is the necessity to counteract excessive inflammation by IL-4 and IL-10?

As shown in Figure 3, the drug treatments had varying effects on IL-6 and TNF-α production in the treated cells. Sorafenib enhanced TNF-α production, while both Acivicin and Azaserine decreased it. In contrast, all three drugs led to an increase in IL-6 production. To further explain the neutral effect of Acivicin and Azaserine on normal breast cells compared to Sorafenib, it was necessary to assess their influence on the production of IL-4 and IL-10, which are anti-inflammatory cytokines. This helped clarify the differential effects of the drugs.

I may agree that, as Abdesttar S et al. state, “ the shift toward anti-inflammatory cytokine production could play a role in increasing immune system recognition of tumor cells, potentially making it more sufficient for the immune system to mount an effective anti-tumor response [citation 21 in tet]. On the other hand, certain anti-inflammatory cytokines could suppress tumor growth by promoting a less permissive tumor environment [citation 22 in text]”, still the eventual relationship among glutamine synthase expression and the different interleukins would be worth of some discussion, at least as author’s advice.

Suggestions.

I suggest that authors would provide some information about uses and documented activities of those molecules, including eventual toxicities (g.e.: about toxicity of Azaserine: doi: 10.1111/apm.13498).

We appreciate the reviewer’s suggestion and have expanded the discussion section to include more details on the biological activities of Acivicin and Azaserine, incorporating the recommended reference.

Few notes:

-in Fig.3A: IL-6 is or is not blunted?

 In Figure 3A, IL-6 levels in MCF-7 cells treated with 6.25 µM of each individual drug gradually increase over time, following a similar pattern. In contrast, the control group treated with DMSO shows a constant level of IL-6 production throughout the treatment period.

-in page 4:

Nexavar treatment resulted in significant cell death, with almost 50% of cells undergoing apoptosis or necrosis, as evidenced by Annexin V staining. This suggests that Nexavar has potent cytotoxic effect

on MCF-7 cells, likely triggering both early and late stages of apoptosis, as well as necrosis. Acivicin induced early apoptotic signaling in over 40% of the treated cells. This indicates that Acivicin primarily promotes early apoptosis without causing as much necrosis or late-stage apoptosis. Azaserine caused a shift toward late-stage apoptosis in about 25% of the treated cells, with a comparable percentage of dead cells (around 15%) to Acivicin.

According to 4.3. Proliferation assay Cytotoxicity, if I understood correctly, cells were incubated for 48 plus 6 hours for MTT evaluation. Thus, after 48 h of incubations with the said molecules cytotoxicity was successful in 50% of Sorafenib treated cells, 40% of Azaserine and 25% of Acivicin treated cells. Is this understanding correct?

Thank you for the insightful conclusion regarding the effects of the indicated drugs on the MCF cell line. However, I would like to clarify the experimental procedure. The cells were first incubated without treatment for 24 hours to allow them to culture in the plates. Following this, the cells were exposed to the specified concentrations of each drug for an additional 24 hours. This procedure was used for the MTT assay and LDH production. The same time schedule was applied in other experiments, where the cells were cultured in 6-well plates. The only difference in these experiments was that the cells were treated with 6.25 µM of each drug.

- in page 7:

Our findings revealed an increasing level of intracellular glutamine in both the control-treated MCF-7

cells and the cells pretreated with Nexavar (Figure 4D). Interestingly, the intracellular glutamine concentration increased (misprint for: not increased ?) in a time-dependent manner in response to Acivicin and Azaserine treatment (Figure 4D). These results suggest that both Acivicin and Azaserine effectively modulate total GS in MCF-7 cells through direct interaction with glutamine synthetase.

Thank you for the reviewer’s feedback. We apologize for the mistake and have corrected the text. The glutamine concentration increased in control-treated cells over time, while it significantly decreased in cells treated with either Acivicin or Azaserine due to their effects on GS activity.

Reviewer 2 Report

Comments and Suggestions for Authors

The article entitled "Effective Targeting of Glutamine Synthetase by Amino Acid Analogs as a Novel Therapeutic Approach in Breast Cancer” aims to investigate the anticancer properties of acivicin and azaserin in the breast cancer cell line MCF-7 and to compare their effects with those on the non-tumorigenic epithelial cell line MCF-10 in vitro.

Can the authors explain the rationale for using the specific concentration range (3.125–50 µM) for treatment with acivicin and azaserin? How do these concentrations compare with physiologically relevant doses in the clinical setting?

Why was Nexavar chosen as a comparator and how does its mechanism of action differ or match that of acivicin and azaserin in the context of this study?

The MCF-7 and MCF-10 cell lines were used in the study. Are there other breast cancer subtypes or more representative models (e.g. 3D cultures or patient-derived xenografts) that could confirm the results?

The study concludes that acivicin and azaserin promote early and late apoptosis with minimal cytotoxicity to normal cells. Could the authors explain why these compounds had different effects on apoptosis stages compared to Nexavar? Is there any evidence of off-target effects influencing these results?

In the docking studies, high binding affinities were observed between the analogs and glutamine synthetase. Were control ligands or inhibitors included to validate the docking predictions?

The authors noted increased levels of IL-4 and IL-10 as a possible anti-inflammatory response. Could alternative explanations, such as cellular stress or compensatory mechanisms, account for these observations?

The authors used LDH release to assess cytotoxicity. Could complementary assays, such as real-time impedance monitoring or live dead staining, substantiate the claims of cytotoxicity?

In Figure 4, intracellular glutamine levels were measured over time. To what extent do the observed changes correlate with specific metabolic rewiring of cancer cells or treatment-induced effects?

The study mentions the use of t-tests for significance analysis. Were corrections made for multiple comparisons to avoid type I errors?

Could the authors provide raw data or additional material to check the reproducibility of the main results, especially for the apoptosis and cytokine tests?

Author Response

Reviewers Response

Reviewer 2

The article entitled "Effective Targeting of Glutamine Synthetase by Amino Acid Analogs as a Novel Therapeutic Approach in Breast Cancer” aims to investigate the anticancer properties of acivicin and azaserin in the breast cancer cell line MCF-7 and to compare their effects with those on the non-tumorigenic epithelial cell line MCF-10 in vitro.

Can the authors explain the rationale for using the specific concentration range (3.125–50 µM) for treatment with acivicin and azaserin? How do these concentrations compare with physiologically relevant doses in the clinical setting?

The provided concentration range was based on previous studies that identified the 50% cytotoxic concentration (IC50) of each drug in different cell lines. For example, Acivicin has an IC50 of 5.4 μM for ALDH4A1 and inhibits the growth of the HepG2 cell line, while Azaserine and Sorafenib have IC50 values around 20 μM. Therefore, we used the recommended concentration ranges to encompass both lethal and safe doses for each drug. However, it is important to note that in vivo and clinical studies operate under different conditions, and we cannot directly correlate these concentrations with those used in such studies. 

https://doi.org/10.1039/C4SC02339K, https://doi.org/10.3390/cancers15174419, https://doi.org/10.1016/0014-4800(90)90044-E

Why was Nexavar chosen as a comparator and how does its mechanism of action differ or match that of acivicin and azaserin in the context of this study?

We selected Nexavar (Sorafenib) as a comparator drug for several reasons. Firstly, Sorafenib is widely used in the treatment of solid tumors, including breast cancer and hepatocellular carcinoma (HCC). As a known Raf-1 inhibitor, Sorafenib's role in regulating proliferation signaling and associated pathways is well established and anticipated. In terms of inhibiting proliferation signaling, Sorafenib is also expected to influence cell cycle arrest and programmed cell death. Therefore, we believed that Sorafenib would serve as an appropriate control for investigating these signaling pathways and mechanisms in the context of drug discovery for new agents.

The MCF-7 and MCF-10 cell lines were used in the study. Are there other breast cancer subtypes or more representative models (e.g. 3D cultures or patient-derived xenografts) that could confirm the results?

All of our findings and hypotheses in the current research have been established using the breast cancer cell line MCF-7 and the normal breast cell line MCF-10. These findings lay the groundwork for future in vivo studies, including patient-derived xenografts and clinical trials.

The study concludes that acivicin and azaserin promote early and late apoptosis with minimal cytotoxicity to normal cells. Could the authors explain why these compounds had different effects on apoptosis stages compared to Nexavar? Is there any evidence of off-target effects influencing these results?

We appreciate the reviewer’s concern, which prompted us to investigate the expression of the tumor suppressor genes PTEN and P53 in treated cells, as shown in Figures 3B and C. Our findings revealed a significant restoration of both PTEN and P53 gene expression in response to Acivicin and Azaserine, but not to Sorafenib. This suggests that Acivicin and Azaserine may have a greater impact on apoptotic signaling compared to Sorafenib.

In the docking studies, high binding affinities were observed between the analogs and glutamine synthetase. Were control ligands or inhibitors included to validate the docking predictions?

Thank you! We have now included the data from the docking studies between GS and its standard inhibitor, Methionine sulfoximine (MS). As a result, a new panel has been added to Figure 6.

The authors noted increased levels of IL-4 and IL-10 as a possible anti-inflammatory response. Could alternative explanations, such as cellular stress or compensatory mechanisms, account for these observations?

We fully agree with the reviewer and have therefore focused on targeting GS through these amino acid’s analogs, considering this mechanism as the primary factor, with all subsequent events being potentially related outcomes.  

The authors used LDH release to assess cytotoxicity. Could complementary assays, such as real-time impedance monitoring or live dead staining, substantiate the claims of cytotoxicity?

We appreciate the reviewer's suggestion. In response, we have generated a new figure (Figure 3) that includes DAPI-stained cells at two different time points: 12 hours and 24 hours post-treatment, as shown in Figure 3A.

In Figure 4, intracellular glutamine levels were measured over time. To what extent do the observed changes correlate with specific metabolic rewiring of cancer cells or treatment-induced effects?

As shown in the new figure (Figure 3), the number of living cells decreased 12 hours after treatment with Acivicin and Azaserine, as indicated by the reduced number of DAPI-stained cells, compared to those treated with Sorafenib or the control. This observation suggests a potential link between the reduction in glutamine concentration over time and its impact on the metabolic processes in cancer cells.    

The study mentions the use of t-tests for significance analysis. Were corrections made for multiple comparisons to avoid type I errors?

In statistical analysis, t-tests are commonly employed to compare up to two groups, usually a control group and a sample group. This method facilitates comparisons between each sample and the control group, enabling separate conclusions to be drawn for each individual drug. As a result, the likelihood of a type I error is generally not anticipated in this type of analysis.

Could the authors provide raw data or additional material to check the reproducibility of the main results, especially for the apoptosis and cytokine tests?

We appreciate the reviewer's concern; however, we have a large dataset that cannot be submitted due to the lack of an option to upload such extensive data on the website. Nonetheless, we have submitted the raw data for the Western blot and confirmed that all of my experiments were repeated three times. Additionally, some of the experiments included three or four different replicates.

Round 2

Reviewer 1 Report

Comments and Suggestions for Authors

Thank you for improving text and understanding of aims of your research.

Reviewer 2 Report

Comments and Suggestions for Authors

The authors responded to my concerns.